# Effects of cardiac contractility modulation on autophagy and apoptosis of cardiac myocytes in rabbits with chronic heart failure

Qingqing Hao [1,2], Shilin Lv [2], Jing Zhang [3], Huiliang Liu [1,2]*

**1** Hebei General Hospital, Shijiazhuang City, Hebei Province, P.R. China, **2** Hebei Medical University, Shijiazhuang City, Hebei Province, P.R. China, **3** Hebei North University, Zhangjiakou City, Hebei Province, P.R. China

* 15030112599@163.com, haoqq2008@hotmail.com

## Abstract

### Background

Cardiac contractility modulation (CCM) is non-excitatory electrical stimulation for improving cardiac function. This study aimed to evaluate the effects of CCM on autophagy and apoptosis of cardiac myocytes in a rabbit model of chronic heart failure (CHF) and explore its possible mechanism.

### Methods

Thirty rabbits were randomised into the Sham, heart failure (HF) and CCM groups, and animals in all three groups were sacrificed after 16 weeks of ascending aortic constriction or sham surgery. The expression of autophagy associated protein LC3 was observed by immunofluorescence staining. With Western-blot measured the expression of Beclin1, P62, LC3B (II/I) and Bcl-2, ALDH2, Bax and Caspase-3 protein in myocardial tissue. The apoptosis rate and the apoptosis of myocardial cells was observed by flow cytometry and TUNEL method.

### Results

1) In comparison to the Sham group, the expression of LC3 and Beclin1 was significantly increased, and the expression of p62 protein was decreased in the heart tissues of rabbits in the HF group. Compared with HF group, after CCM intervention, the expression of Beclin1 and LC3B proteins decreased, while the P62 protein increased, and the LC3B(II/I) ratio decreased (P<0.05). 2) The expression of Bcl-2, ALDH2 protein and Bcl-2 mRNA decreased compared with the Sham group (P<0.05), while the expression of Bax, Caspase-3 protein and mRNA was significantly increased (P<0.05). However, the expression of ALDH2 mRNA in the CCM group was not statistically significant. The expression of Bcl-2, ALDH2 protein and mRNA increased after CCM intervention, and the expression of Bax, Caspase-3 protein and mRNA decreased (P<0.05). 3) The apoptosis situation in the Sham group was similar to that of normal myocardium, compared with the Sham group, the number of apoptotic bodies increased, and the apoptosis percentage of cardiomyocytes

**Data Availability Statement:** All relevant data are within the paper and its Supporting Information files.

**Funding:** The authors received financial support for this research from the following sources: the "Three Three Three Talents Project" of Hebei Province (Grant No. A202101065, https://rst.hebei.gov.cn/); the Medical Science Research Project of Hebei Province (Grant No. 20241645, http://wsjkw.hebei.gov.cn/) awarded to QH; and the Government Funds Programs for Talented People (Grant No. ZF2024019, https://czt.hebei.gov.cn) awarded to HL. The funders had no role in study design, data collection and analysis, decision to publish, or preparation of the manuscript.

**Competing interests:** The authors have declared that no competing interests exist.

increased significantly (P<0.05). After CCM intervention, the number of apoptotic bodies and the percentage of apoptosis decreased compared with the HF group (P<0.05).

## Conclusions

The intervention of CCM has been shown to enhance both myocardial systolic and diastolic function in rabbits with CHF. The mechanism may be related to the inhibition of cardiomyocyte autophagy by regulating the expression levels of Beclin1, P62, and LC3B(II/I) in cardiomyocytes, as well as the reversal of cardiomyocyte apoptosis by regulating the expression levels of Bcl-2, ALDH2, Bax, and Caspase-3 in cardiomyocytes.

## 1. Introduction

Chronic heart failure (CHF) is the terminal stage of various structural and functional cardiac diseases [1]. Despite recent advances in the treatment of CHF, the morbidity and mortality among patients with CHF remain high. The treatment of chronic heart failure is mainly divided into two parts: drug therapy and device therapy. Classic drugs for chronic heart failure include beta-blockers, mineralocorticoid receptor antagonists, renin-angiotensin system inhibitor, sodium-glucose cotransporter 2 inhibitors, ivabradine, and more [2]. Device therapies, including left bundle branch area Pacing, cardiac resynchronization therapy, and ventricular assist devices for HF, have made impressive progress in the last decade. Despite numerous attempts to treat CHF in recent decades, it remains a prevalent and challenging disease. Approximately 30% of patients worldwide suffer from this condition [3].

CCM is the administration of high-intensity electrical stimulation during the absolute refractory period of cardiomyocytes, which does not cause depolarization of cardiomyocytes but the contractility of myocardium, thereby improving cardiac function, and plays an important role in the treatment of chronic heart failure [4, 5].Previous studies have shown that CCM therapy has beneficial effects by regulating calcium handling, the cytoskeleton, the extracellular matrix, and potentially the autonomous nervous system. However, the association between CCM and autophagy and apoptosis has rarely been studied.

Beclin-1, LC3, and P62 are commonly used proteins to indicate the level of autophagy. When the expression of Beclin-1 increases, the autophagy level of cardiac myocytes also increases correspondingly, the amount of LC3II protein or the ratio of LC3II/LC3I is positively correlated with the number of autophagosomes, and the expression of P62 is negatively with the degree of autophagy [6]. Acetaldehyde dehydrogenase 2(ALDH2) and Bcl-2 have anti-apoptotic function, while Bex promotes cell apoptosis [7–9]. In this study, a rabbit model of CHF was established by ascending aorta constriction, and the effect of CCM on autophagy and apoptosis of ventricular muscle in the model was investigated.

## 2. Materials and methods

### 2.1. Animals

Thirty healthy New Zealand white rabbits (6 months old, weight: 2.5–3.5 kg, both male and female rabbits) were provided by the Experimental Animal Center of Hebei Medical University.

### 2.2. Experimental grouping

Thirty rabbits were randomly divided into three groups (n¼10) as follows: the sham operation group (sham group), the HF group (HF group), and the CCM group. Rabbits in the sham

group only received thoracotomy. Thoracotomy and ascending aortic cerclage were performed in the HF group. In the CCM group, rabbits underwent thoracotomy and ascending aortic cerclage, and received 4-week CCM after the formation of CHF.

## 2.3. Establishment of animal model of chronic heart failure

The rabbits in HF and CCM groups were anesthetized via the ear vein with 3% sodium pentobarbital at 1 ml/kg body weight, and their thoracic cavities were opened. Subsequently, their ascending aorta was dissected until the aorta circumference was constricted up to 60%. After 12 weeks, symptoms of heart failure occurred in the rabbits, including appetite reduction, breathing acceleration, and fewer activities. The HF model was thought to be successfully established while the left ventricular ejection fraction reached 40%.

Before the experiment, an electrode was used to deliver CCM. Specifically, one end of the electrode was sutured to the left ventricular anterior wall of rabbits, and the other end was punctured subcutaneously to the neck. Then, a cardiac stimulator (EPS320, BARD Micro-Pace, Inc., USA) was used to deliver CCM signals to the heart by sensed R-wave at the absolute refractory period. These signals consisted of biphasic square-wave pulses with phase duration of 2 MS, stimulus amplitude of 7 V, and 30 MS delay after R-wave sensing [10]. The CCM signals lasted 6 hours per day for a consecutive 4 weeks.

After 16 weeks post ascending aortic constriction (AAC) or sham surgery, the animals in each group were euthanized with an overdose of anesthesia. The hearts were excised and flushed with physiological saline to remove any residual blood from the heart chambers. The surface liquid of the hearts was blotted dry with filter paper, and the heart weights were recorded. Myocardial tissue from the anterior wall of the left ventricle was collected using ophthalmic scissors and fixed in 4% paraformaldehyde for one day for immunofluorescence staining. The tissue underwent routine dehydration with a graded alcohol series (70%, 80%, 90%, 100%), clearing with xylene (three times), wax infiltration at 65˚C, embedding in paraffin blocks, and sectioning into 4μm consecutive slices.

## 2.4. Autophagy associated protein LC3 immunofluorescence staining

The slices were stored in refrigerator at 4˚C. After rewarming, the slices were baked at 60˚C for 30min, and then routinely were dewaxed to water. After antigen repair, it was placed in the goat serum blocking solution and incubated at 37˚C for 30 minutes. The test tablet was dripped with a properly diluted primary antibody (Hebei Bohai Biological Engineering Company) stayed at 4˚C overnight, and rewarmed at 37˚C for 30min after taking out. The secondary antibody (DyLight488 Fluorescein-labeled Sheep Anti-Rabbit IgG produced by Hebei Bohai Biological Engineering Company) was dropped at 37˚C and incubated at 37˚C for 45 minutes. The experimental sections were re-stained with DAPI staining solution (Hebei Bohai Biological Engineering Company) and incubated at room temperature for 3min. The tablets were sealed with anti-fluorescence quenched tablets. Observe and take photos under fluorescence microscope. (Image-Pro Plus produced by American Media Cybernetics, Inc)

## 2.5. Western blot to detect the expression level of Beclin1, P62, LC3B (II/I) protein

The frozen myocardium of 0.1g was weighed, split and centrifuged, and the frozen myocardium of 0.1g was weighed, split and centrifuged, and its protein content was determined by BCA method. According to the standard of 30μg protein per sample hole, the sample amount of each lane was calculated, and the sample was added to the prepared SDS-PAGE electrophoresis gel. The pre-dyed marker (Boster Biological Technology) was added to the second lane

on the left, and the loading buffer was added to the empty lane. The sealing solution was prepared by adding 2.5g of 5% calf serum (Sigma-Aldrich) to TBST. Through SDS-PAGE electrophoresis, the protein on the gel was transferred to the PVDF film and immersed in calf serum sealing solution. At 37˚C, it was horizontally shaken and sealed for 1h. The primary antibody was diluted by sealing solution: Beclin1 (1:1000), P62 (1:1000), LC3B (1:1000) (Cell Signaling Technology). The closed PVDF membrane was removed and the TTBS was washed three times. PVDF membrane was transferred to the reaction box, appropriate amount of primary antibody was added, 4˚C, horizontal oscillation, incubated overnight. Take out the PVDF membrane treated with primary antibody, rewarm it to room temperature naturally, and rinse it with TTBS oscillating for 3 times, 15min each time. PVDF membrane was transferred to the reaction box, secondary antibody was added (secondary antibody was diluted by TTBS, Beclin1 (1:2000), P62 (1:2000), LC3B (1:2000). At 37˚C, it was horizontally shaken and incubated for 2h. After that, the film was washed with TTBS at room temperature. The luminous solution was dropped on the PVDF film, observed the film with an imager and photographed strips. ProPlus5.1 software was used for scanning and analyzing the gray value of the strip, and the relative expression level of target protein was expressed by the gray ratio of target strip to β-action strip.

## 2.6. Analysis of mRNA expression of Bcl-2, Bex, Caepase-3 and LADH2 by RT-PCR

The 0.1g frozen myocardial tissue was weighed, ground and centrifuged, and the purity and integrity of RNA were detected by 1% agarose gel electrophoresis and type 756 ultraviolet spectrophotometer, it was reverse-transcribed at 42˚C for 50min and reverse transcriptase was inactivated at 95˚C for 5 min. The PCR thermal cycle is carried out: 96˚C for 4 min, followed by three steps of reaction: 94˚C for 30 s, 58˚C for 30 s, and 72˚C for 30 s. Fluorescence signals were collected at the third step of each cycle and detected by Syber Green fluorescent quantitative PCR. (The fluorescence quantitative RT-PCR kit was made from Promega Company, USA) After amplification, GAPDH was used as the internal reference gene for result analysis. Compared with the control group, the relative quantitative value (RQ value) of target gene expression was obtained, and the RQ value was used for statistical analysis.

The primer sequences of the genes used in the RT-PCR (Table 1).

## 2.7. Flow cytometry

The cells of each group were inoculated into 6-well plates at a density of 1×105 cells/well and cultured for 48 h. The cells are collected after trypsin digestion. After 3 times of washing with

**Table 1. The primer sequences of the genes used in the RT-PCR.**

| Gene | Primer Sequence(5'-3') | Length(base pair) |
|---|---|---|
| GAPDH | Forward: TGAACGGGAAGCTCACTGG | 120 |
| | Reverse: GCTTCACCACCTTCTTGATGTC | |
| Bcl-2 | Forward: GCCTTCTTTGAGTTCGGTG | 143 |
| | Reverse: ACCCAGCCTCCATTATCCT | |
| Bax | Forward: CGACGGCAACTTCAACTG | 114 |
| | Reverse: CAGCCCATAATAGTCCTGATGA | |
| Caspase-3 | Forward: AAGCCACGGTGATGAAGGA | 136 |
| | Reverse: TCGGCAAGCCTGAATAATG | |
| LADH2 | Forward: TGGACAAAGTGGCGTTCA | 78 |
| | Reverse: TCTTGAGGTTACTGCTCCCTG | |

PBS, the cells were re-suspended in 100 μL binding buffer and mixed with Annexin V-FITC (5 μL) and PI (10 μL) dye solution, dying at room temperature and away from light for 15 min, the apoptosis of cardiomyocytes was detected with Annexin V-FITC/PI apoptosis detection kit. Flow cytometry produced by BD Biosciences in the United States was used to detect according to the operating instructions.

## 2.8. Detection of cardiomyocyte apoptosis

The myocardium tissue was cut, dehydrated and immersed in wax, frozen and buried, and then cut into 3μm slices, then spread the slices in water bath and scoop the slices. The slices were heated for 1h in a constant temperature oven at 65˚C, soaked in xylene for 30min, dehydrated with gradient alcohol and rinsed with tap water for 10min. The pepsin at 0.5% concentration was incubated at 37˚C for 15min, and 50μL Tunel reaction mixture was added to the sample, and the slices were placed in a humidor and incubated at 37˚C for 1h. 50μL converted POD was added and incubated at 37˚C for 30min. 50μL DAB substrate was added and incubated at room temperature for 10min. Hematoxylin was re-dyed, dehydrated, transparent and photographed. The Tunel kit produced by Roche was used to detect following the operating instructions. The apoptotic bodies were observed by optical microscope and the images were preserved.

## 2.9. Statistical methods

All data were analyzed and processed with SPSS19.0 software. The measurement data is expressed by mean±standard deviation. T-test is used to compare the mean of the two samples when the distribution is normal and the variance is homogeneous, and the corrected t test is used when the variance is not homogeneous; the rank sum test is performed for non-normal distribution. Single factor analysis of variance was used to compare the mean of multiple samples, and then SNK-q method was used to compare the two samples, $P < 0.05$ was statistically significant.

# 3. Results

## 3.1. Establishment of the HF model

There were no deaths in the sham group during surgery. One rabbit died during surgery owing to pneumothorax in the CHF group and one rabbit died as a result of artery rupture in the CCM group. Echocardiography confirmed that the HF model was successfully established.

## 3.2. Autophagy associated protein LC3 immunofluorescence staining

The results of immunofluorescence staining showed that the myocardial tissue of the sham-operated group had a basic amount of LC3 immunofluorescence positive staining, showing red fluorescent spots. Compared with the sham-operated group, the density of LC3 immuno-fluorescence staining spots in the myocardial tissue of rabbits in the heart failure group increased, suggesting that autophagy activity of myocardial cells in the animals increased after heart failure; Compared with the heart failure group, the LC3 immunofluorescence staining in the myocardial tissue of animals after CCM intervention decreased, and the density of positive fluorescent spots decreased. It suggested that CCM intervention inhibits the enhancement of autophagy activity in heart failure animals (Fig 1).

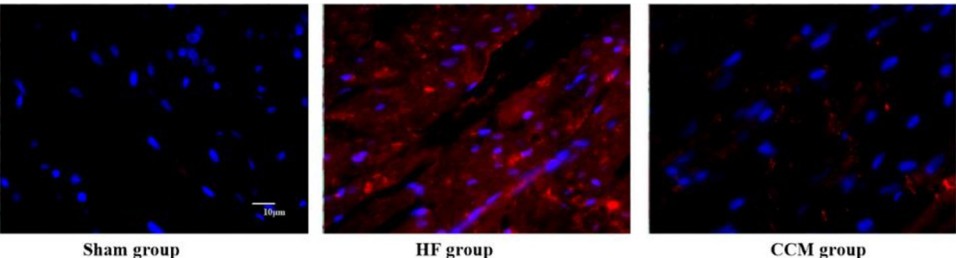

**Fig 1. Comparison of myocardial LC3 immunofluorescence staining of three groups (×400, scale bar = 10μm).**

## 3.3. Western blot to detect the expression level of Beclin1, P62, LC3B (II/I) protein

Beclin1, P62, and LC3B proteins in the myocardium of animals in the sham-operated group had a certain amount of basic expression; Compared with the sham-operated group, the expression of Beclin1 protein was significantly increased, the expression of P62 protein was decreased, the expression of LC3B protein was significantly increased, and the ratio of LC3B (II/I) was increased in heart failure group ($P < 0.05$). It suggests that CHF leads to increased autophagy activity of cardiac myocytes. After CCM intervention, compared with HF group, the expression level of Beclin1 protein in myocardial tissue was significantly decreased, the expression level of P62 protein was significantly increased, the expression of LC3B protein was significantly decreased, and the ratio of LC3B (II/I) was decreased ($P < 0.05$), it is suggested that CCM intervention can effectively inhibit the increase of autophagy activity of cardiac myocytes caused by heart failure (Fig 2 and Table 2).

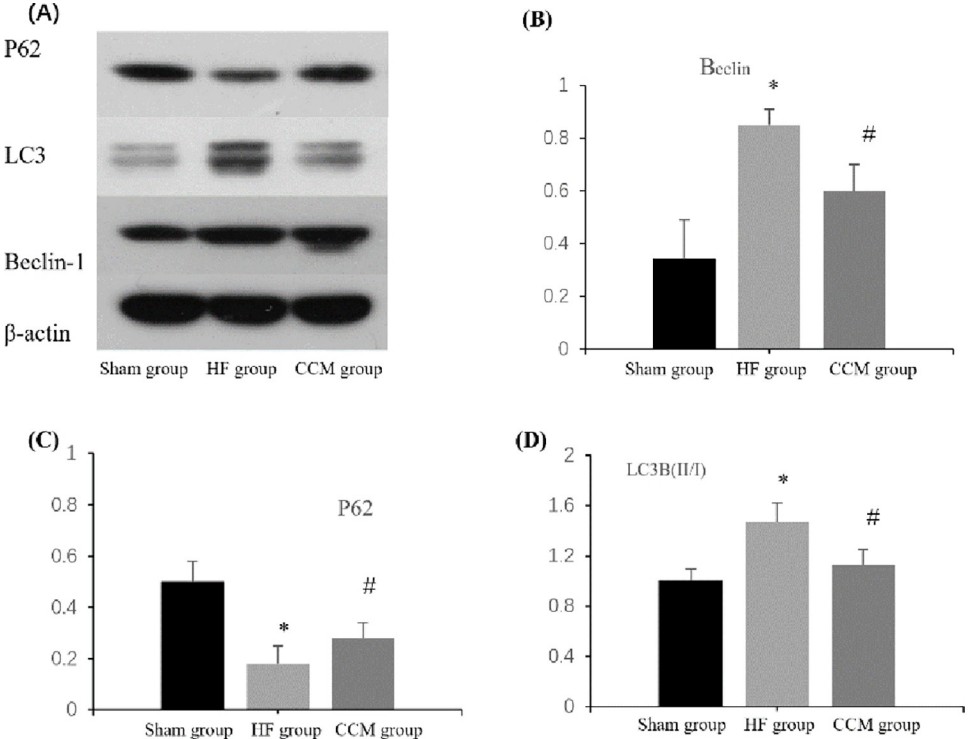

**Fig 2.** (A)Western blot images of Beclin1, P62, LC3B(II/I) in three groups; (B) Beclin1expression; (C) P62 expression; (D) LC3B(II/I) expression. Note: *P < 0.05 vs sham group; #P < 0.05 vs HF group The columns and error bars represent means ±SD.

**Table 2. Comparison of Beclin1, P62 and LC3B(II/I) protein expression levels in three groups.**

|  | Sham group | HF group | CCM group |
|---|---|---|---|
| Beclin1 | 0.34±0.15 | 0.85±0.06△ | 0.60±0.11△▲ |
| P62 | 0.50±0.08 | 0.18±0.07△ | 0.28±0.06△▲ |
| LC3B(II/I) | 1.00±0.10 | 1.47±0.15△ | 1.13±0.12▲ |

Note

△$P < 0.05$ vs sham group

▲$P < 0.05$ vs HF group.

### 3.4. Analysis of mRNA expression of Bcl-2, Bax, Caspase-3 and LADH2 by RT-PCR

There were significant differences in the expression of Bcl-2, Bax mRNA (GAPDH as the internal reference gene), LADH2 and Caspase-3 mRNA among the three groups. The expression of Bcl-2, Bax mRNA and Caspase-3 mRNA in HF group and CCM group was significantly lower than that in SHAM group ($P<0.05$), and the expression of Bax and Caspase-3 mRNA was significantly higher ($P<0.05$). The expression of ALDH2 mRNA in HF group was significantly lower than that in SHAM group ($P<0.05$), while the decrease of ALDH2 mRNA expression in CCM group was not statistically significant. After CCM intervention, the expression of Bcl-2 and ALDH2 mRNA in CCM group was significantly higher than that in HF group, while the expression of Bax and Caspase-3 mRNA was significantly lower ($P<0.05$). The ratio of Bcl-2 /Bax in the three groups was $0.92 \pm 0.22$, $0.05 \pm 0.14$ and $0.17 \pm 0.04$ respectively. After CCM stimulation, the ratio of Bcl-2/Bax in CCM group was significantly higher than that in HF group (Fig 3 and Table 3).

### 3.5. Observation of cardiomyocyte apoptosis by flow cytometry

The study found significant differences in the changes of cardiomyocyte apoptosis among the three groups. The rate of cardiomyocyte apoptosis in the HF group was significantly higher than that in the Sham group ($P<0.05$). Whereas the rate of cardiomyocyte apoptosis in the CCM group was significantly lower than that in the HF group ($P<0.05$) (Figs 4–7 and Table 4).

### 3.6. Observation of cardiomyocyte apoptosis by TUNEL method

The apoptosis of the Sham group was similar to that of the normal myocardium. The number of apoptotic bodies in the HF group and CCM group increased compared with that in the SHAM group, while the number of apoptotic bodies in the CCM group decreased compared with that in the HF group (Fig 8).

## 4. Discussion

Chronic heart failure (CHF) is a clinical syndrome of abnormal cardiac structure and (or) function caused by decompensation of myocardial tissue to a variety of pathogenic factors, which is easily complicated by malignant ventricular arrhythmia, and is the main cause of death. Some studies have shown that autophagy accelerates heart muscle cell death in the progression to heart failure, exacerbating heart failure [11, 12]. Cardiomyocyte apoptosis is an important factor in the process of central chamber remodeling and cardiac function deterioration. By inhibiting autophagy and apoptosis of cardiomyocytes, it can effectively improve heart function, delay the occurrence and development of heart failure, and achieve the purpose

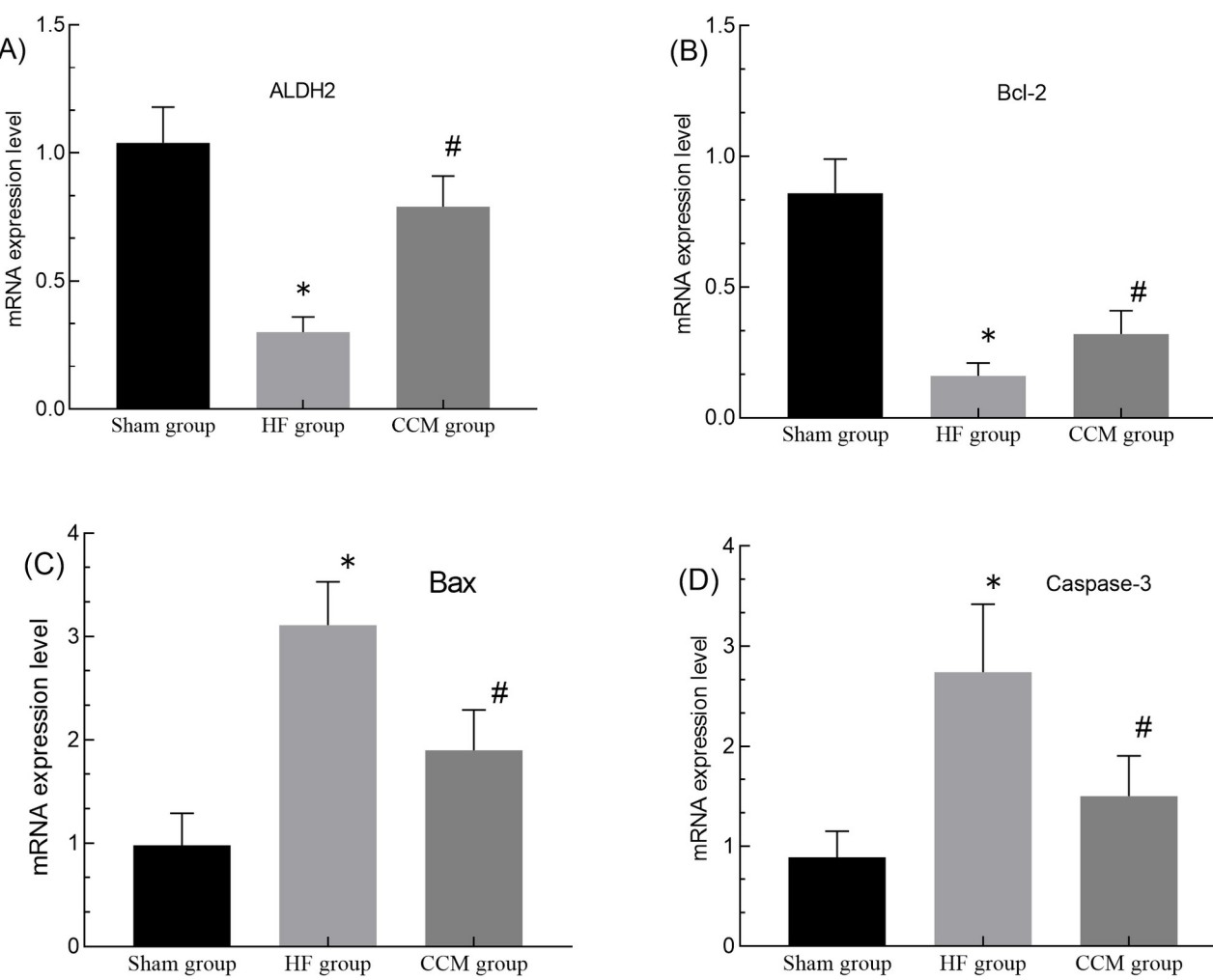

**Fig 3.** (A) Expression of ALDH2 mRNA in heart tissues (fluorescence quantitative RT-PCR). (B) Bcl-2 mRNA. (C) Bax mRNA. (D) Caspase-3 mRNA. Note: *P < 0.05, HF group vs SHAM group. #P < 0.05, CCM group vs HF group.

of prevention and treatment of heart failure. Rabbit hearts have some physiological similarities with human hearts, so we selected a rabbit CHF model induced by aortic coarctation to study the effects of CCM on autophagy and apoptosis of cardiomyocytes in rabbits with chronic

**Table 3. Changes in mRNA expression of Caspase-3, ALDH2, Bcl-2, Bax and Bcl-2/Bax in three groups.**

|           | Sham group | HF group | CCM group |
|-----------|-----------|----------|-----------|
| ALDH2     | 1.04±0.14 | 0.30±0.06△ | 0.79±0.12▲ |
| Bcl-2     | 0.86±0.13 | 0.16±0.05△ | 0.32±0.09△▲ |
| Bax       | 0.98±0.31 | 3.11±0.42△ | 1.90±0.39△▲ |
| Caspase-3 | 0.89±0.26 | 2.74±0.68△ | 1.47±0.35△ |
| Bcl-2/Bax | 0.92±0.22 | 0.05±0.14△ | 0.17±0.04△▲ |

Note

△P < 0.05 vs sham group

▲P < 0.05 vs HF group.

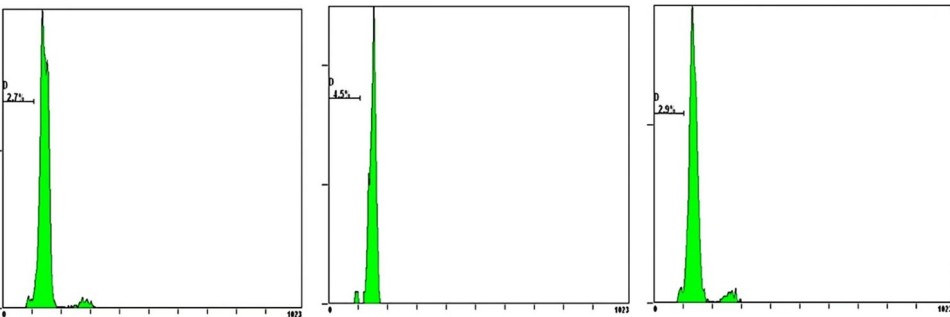

**Fig 4. Cardiomyocyte apoptosis rate in the Sham group.**

heart failure. The results show that: CCM intervention can improve the systolic and diastolic function of myocardium in rabbits with chronic heart failure [13–16]. The mechanism may be related to inhibiting autophagy of cardiomyocytes by regulating the expression levels of Beclin1, P62 and LC3B(II/I), and reversing the development of apoptosis of cardiomyocytes by regulating the mRNA and protein expression levels of Bcl-2, ALDH2, Bax and Caspase-3 in cardiomyocytes.

Autophagy plays an important role in ventricular remodeling. It has been reported that [17] autophagy can aggravate the ventricular remodeling induced by pressure load and affect cardiac function. Lots of proteins are involved in autophagy, many of which have been widely studied as commonly used autophagy marker factors. Beclin-1, LC3 and P62 are commonly used index proteins to reflect the level of autophagy. When the expression of Beclin-1 increases, the autophagy level of cardiac myocytes will also increase correspondingly, and pathological ventricular remodeling will accelerate. On the contrary, after the deletion of Beclin-1, the expression of this gene will be lost, and the autophagy level of cardiac myocytes will decrease, reducing the occurrence of ventricular remodeling. LC3 is an essential ubiquitin-like pathway key protein in the process of autophagy. It participates in the formation of autophagosomes and is an indispensable molecule in the formation of autophagolysosomes. The content of LC3II protein or the ratio of LC3II/LC3I were positively correlated with the number of autophagosomes, which could reflect the activity of autophagy to a certain extent [18]. Autophagy degradation substrate P62 interacts with LC3 through LC3 recognition sequence and trantranzes ubiquitination protein to autophagosome, degrading the degraded substance in the autophagolysosome, and eliminating P62 along with the clearance of the polymer [19]. When the autophagy activity of the cell increases, the expression of P62 protein decreases, showing a negative correlation with the degree of autophagy [6]. After 12 weeks, the

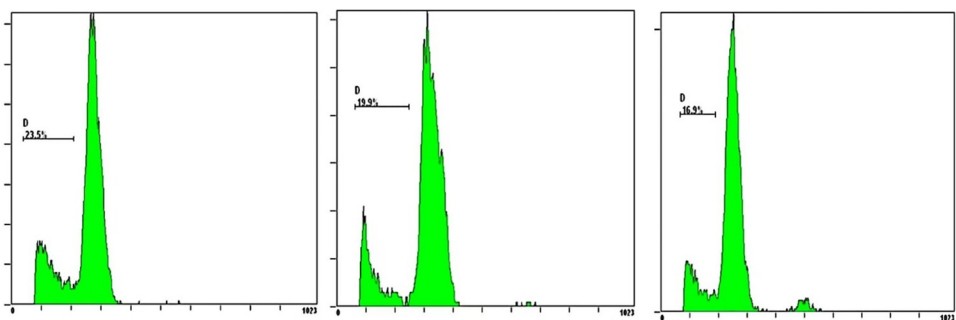

**Fig 5. Cardiomyocyte apoptosis rate in the HF group.**

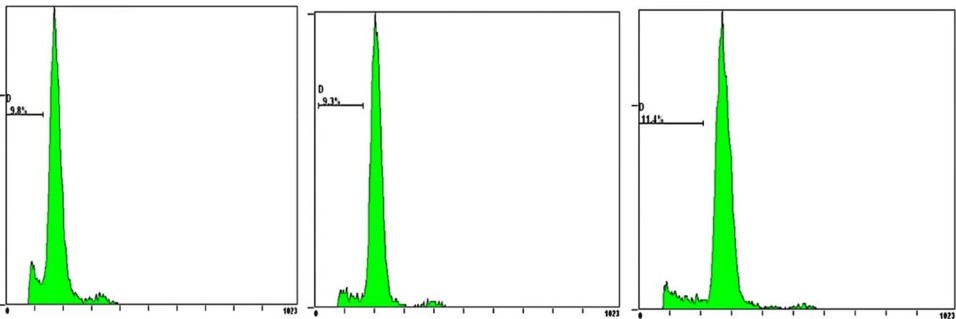

**Fig 6. Cardiomyocyte apoptosis rate in the CCM group.**

heart function of the animal was in the decompensated period. Compared with the sham operation group, the expression of Beclin1 and LC3B (II/I) protein in the heart failure group was significantly increased ($P<0.05$), and the expression of P62 protein was decreased ($P<0.05$), suggesting that the autophagy level of the animal cardiac myocytes was increased during heart failure, which was similar to the above research results. After CCM treatment, the expression of Beclin1 and LC3B (II/I) protein in myocardial tissue of heart failure animals decreased ($P<0.05$), and the expression of P62 protein increased ($P<0.05$), suggesting that the improvement of cardiac function of heart failure animals by CCM may be related to the inhibition of autophagy of cardiac cells by CCM.

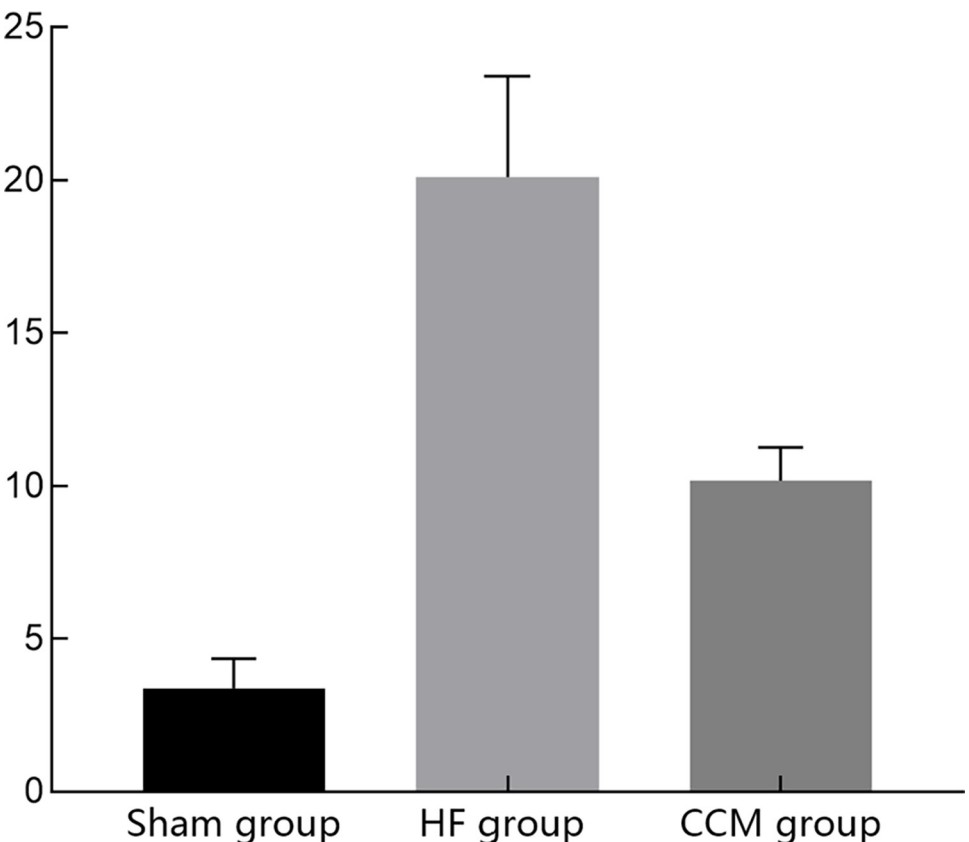

**Fig 7. Cardiomyocyte apoptosis rate in the three groups.**

**Table 4. Apoptosis ratio in three groups.**

|  | Sham group | HF group | CCM group |
|---|---|---|---|
| Apoptosis (%) | 3.37±0.98 | 20.1±3.30△ | 10.17±1.09△▲ |

Note

△P < 0.05 vs sham group

▲P < 0.05 vs HF group

Through the study of autophagy and heart failure, it is found that autophagy plays an important role in the pathophysiological process of heart failure. In this study, it was found that while CCM stimulation improved cardiac function, autophagy activity of cardiomyocytes was weakened, suggesting that the effect of CCM on cardiac function in animals with chronic heart failure may be related to autophagy, and its mechanism needs to be further studied.

Apoptosis is an active death process that is generally regulated by multiple factors in the body [20]. It plays an important role in the pathogenesis of heart failure. ALDH2 is widely distributed in human body, which has anti-apoptotic function and plays a role in cardiac protection. ALDH2 is mainly present in mitochondria, but is especially expressed in the heart, and the ALDH2 gene is down-regulated in hypoxic myocardium. BCL2 family members play a crucial role in the regulation of apoptosis. Bcl-2 and Bax are the two most important members of BCL2 family members involved in apoptosis regulation. Bcl-2 can promote cell survival and play an anti-apoptotic role. BCL2 family members play a crucial role in the regulation of apoptosis. Bcl-2 and Bax are the two most important members of BCL2 family members involved in apoptosis regulation. Bcl-2 can promote cell survival and play an anti-apoptotic role [8, 9, 21].

This study found that the expression of Bcl-2, ALDH2 mRNA in the heart failure group was significantly lower than that in the sham operation group (P<0.05), and the expression of Bax, Caspase-3 mRNA was significantly higher (P<0.05), suggesting that the apoptosis activity of myocardial cells in the heart failure group was increased. After CCM intervention, the expression of Bcl-2, ALDH2 mRNA in myocardial tissue of animals was significantly increased, while the expression of Bax, Caspase-3 mRNA was significantly decreased (P<0.05). It is suggested that the improvement of cardiac function by CCM may be related to the inhibition of cardiomyocyte apoptosis by CCM. Its mechanism is relatively complex, and it may be related through various ways, such as inhibiting oxidative stress response, reducing Ca2

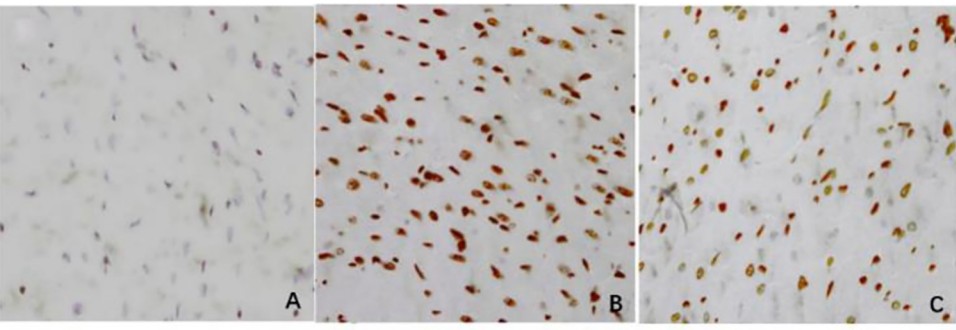

A.Sham group     B.HF group     C.CCM group

**Fig 8. Tunel detection of myocardial apoptosis.**

+ concentration in the periphery of the sarcoplasmic reticulum calcium pump, enhancing the expression of myocardial protective factors, and so on, which needs further research to confirm.

This study shows that CCM can improve cardiac function in rabbits with chronic heart failure, which may be related to inhibiting autophagy and apoptosis of myocardium. However, the specific mechanism of the effect of cardiac contractile force regulation on autophagy and apoptosis of cardiomyocytes in rabbits with chronic heart failure needs further study.

At present, the treatment of heart failure mainly includes drug therapy and non-drug therapy, although in recent decades, the drug and device treatment of heart failure has been greatly improved, the 5-year survival rate of heart failure patients is still about 50%, and the mortality rate remains high [22, 23]. The traditional "golden triangle drugs" (β-receptor blocker, angiotensin-converting enzyme inhibitors or angiotensin receptor blockers, aldosterone antagonists) have been the main drugs in clinical treatment of heart failure in the past two decades. These drugs have played a significant role in reducing the death rate from heart failure [24]. Since the 1990s, the use of ivabradine and sacubactril/valsartan has provided more drug options for the clinical treatment of heart failure [25, 26]. In terms of device therapy, implantable cardioverter defibrillators (ICDs), biventricular cardiac resynchronization therapy (CRT) and ventricular assist devices have also been gradually used to treat heart failure in the past few decades, which has brought great help to heart failure patients [27]. However, there are also many limitations, such as ICDs can extend the survival of patients with heart failure, but do not improve heart function and symptoms. Many patients with heart failure do not have an indication for CRT or VAD implantation because the QRS duration is not extended enough or their symptoms are not severe enough. Therefore, there is still a need to further explore new and effective treatments.

Some limitations of this study should be noted. First, the sample size was relatively small. Second, HF is a complex process, and it is unclear whether other models of HF, such as volume overload or tachycardia, lead to similar outcomes.

In conclusion, autophagy and apoptosis play important roles in the pathophysiology of heart failure, Chronic heart failure results in enhanced autophagy and apoptosis of cardiomyocytes. These results provide experimental basis for the application of CCM in the treatment of CHF. The results of this study indicate that CCM can improve the cardiac function of rabbits with chronic heart failure, which may be related to the inhibition of myocardial autophagy and apoptosis. However, the specific mechanism of the effect of cardiac contractile force regulation on autophagy and apoptosis of cardiomyocytes in rabbits with chronic heart failure needs further study. Current clinical studies have confirmed that exercise tolerance and quality of life of patients with heart failure are improved after CCM application. But there is still a lack of large-scale clinical trials to confirm the long-term safety of CCM in the treatment of heart failure, and there is also a lack of large-scale randomized controlled studies on the effects of CCM on autophagy and apoptosis of cardiomyocytes in chronic heart failure. Therefore, it is expected to further study the effect of CCM on ventricular muscle autophagy and apoptosis and its mechanism, so as to provide ideas for the treatment of heart failure.

## Supporting information

**S1 File.**
(XLSX)

**S1 Raw images.**
(PDF)

## Author Contributions

**Data curation:** Qingqing Hao, Jing Zhang, Huiliang Liu.

**Methodology:** Qingqing Hao, Shilin Lv.

**Project administration:** Qingqing Hao.

**Software:** Shilin Lv, Huiliang Liu.

**Supervision:** Huiliang Liu.

**Writing – original draft:** Qingqing Hao, Shilin Lv.

**Writing – review & editing:** Qingqing Hao, Shilin Lv, Jing Zhang.

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
