## [Decision Letter · Decision Letter 0]

13 Dec 2023

PONE-D-23-21385Effects of cardiac contractility modulation on autophagy and apoptosis of cardiac myocytes in rabbits with chronic heart failurePLOS ONE

Dear Dr. Hao,

Thank you for submitting your manuscript to PLOS ONE. After careful consideration, we feel that it has merit but does not fully meet PLOS ONE’s publication criteria as it currently stands. Therefore, we invite you to submit a revised version of the manuscript that addresses the points raised during the review process.

Your manuscript was reviewed by two experts and we received positive feed with major comments.

We look forward to receiving your revised manuscript.

Kind regards,

Partha Mukhopadhyay, Ph.D.

Section Editor

PLOS ONE

Journal Requirements:

3. To comply with PLOS ONE submissions requirements, in your Methods section, please provide additional information regarding the experiments involving animals and ensure you have included details on (1) methods of sacrifice, and (2) efforts to alleviate suffering.

6. Please include a copy of Table 1-3 which you refer to in your text

Reviewers' comments:

Reviewer's Responses to Questions

**Comments to the Author**

1. Is the manuscript technically sound, and do the data support the conclusions?

Reviewer #1: Yes

Reviewer #2: Partly

2. Has the statistical analysis been performed appropriately and rigorously? 

Reviewer #1: Yes

Reviewer #2: Yes

3. Have the authors made all data underlying the findings in their manuscript fully available?

Reviewer #1: No

Reviewer #2: No

4. Is the manuscript presented in an intelligible fashion and written in standard English?

Reviewer #1: Yes

Reviewer #2: No

5. Review Comments to the Author

Reviewer #1: This manuscript by Hao et al. reports an interesting study on the application of cardiac contractility modulation (CCM) as an instrument treatment method in case of chronic heart failure. They showed that during heart failure in the rabbit animal model, CCM treatment can be implemented to improve/restore the systolic and diastolic behavior of the heart tissue (myocardium), which might be the result of the inhibition of the cardiomyocytes autophagy. The study itself is well-designed and executed, however, the manuscript is written with little information about the background information on this field and the discussion lacks proper justification/clarification of the presented data. Below are the comments:

1. Authors provide a very short description of the chronic heart failure (CHF) and also mention the current/previous treatment methods (mostly beta-blocker) that have been attempted to treat this disease. However, for the general reader, it would be quite challenging to understand what is CHF and what happens during this phase. Moreover, it would be beneficial for the readers to briefly state the mode of action of the beta-blocker and how/why this method of treatment is less-effective (if it is). Why do the authors think that CCM can be a better strategy to treat CHF?

2. In the line 45-47, authors used a term “molecular modeling”. What does it mean? Molecular modeling of what? How is it happening? (Sorry, if I missed that info somehow!).

3. The methods section is well-written. However, materials section requires more details about the reagents and instruments that were used in these experiments.

4. The line 79 starts with “The slices were stored...”. Slices of what?

5. In figure 3, the authors checked the expression level of Bcl-2, Bax, Caspase-3 and LADH2 in Sham, HF and CCM groups and concluded that the expression of these mRNAs and the respective proteins in cardiomyocytes can reverse the development of cardiomyocytes apoptosis. Can authors also check the protein levels (by western blotting) from those differently-expressed mRNA in Sham, HF and CCM groups and validate the above-mentioned statement?

6. Section 3.6 (line 176-180), authors looked at the cardiomyocytes apoptosis level by TUNEL assay, where they found that overall the apoptosis rate in HF and CCM group was increased compared with that in Sham group. Do authors have any justification of this observation? Also, is there any way to quantify this data (including the statistical significance) to get a better analysis of the apoptosis level?

7. Where is table 1, table 2 and table 3 listed in the manuscript? (Mentioned after the figure legend of figure 2, figure 3 and figure 6).

8. Discussion requires more details and proper explanation of the results that authors presented in the manuscript. What is the molecular mechanism behind the effectiveness of CCM treatment in case of chronic heart failure? How does up regulation/down regulation of those proteins tested here in CCM group of animals justify this treatment mechanistically during chronic heart failure? In the discussion section, authors mentioned that previously they showed the positive effect of CCM in cardiac function, however, did not cite any of their publication. Why?

Overall, this study has merit and the experiments are well controlled and can be a nice addition to the research community of chronic heart failure. That being said, the manuscript requires additional details on the background, significance of this study and proper explanation of their data.

Reviewer #2: There are a few major corrections that need to be met before acceptance of manuscript could be made specifically in materials and methods section, results section. The overall language of the manuscript needs editing too, in order to match the standards of scientific reporting and the journal.

6. PLOS authors have the option to publish the peer review history of their article (what does this mean?). If published, this will include your full peer review and any attached files.

Reviewer #1: No

Reviewer #2: No

---

## [Author Response · Author response to Decision Letter 0]

14 Apr 2024

Thank you for raising your concerns, and we have taken diligent steps to address them. We have thoroughly revised the manuscript and uploaded the revised version, along with the modified manuscript, to the designated location as per your instructions. Additionally, we have provided a separate Word document containing all the figures and tables, ensuring clarity and convenience. To further enhance transparency and completeness, we have included documentation confirming ethical approval from the ethics committee.

During the revision process, we meticulously attended to each of your requirements and conducted a comprehensive review of the manuscript to ensure accuracy and consistency across all sections. Your valuable suggestions have been carefully incorporated, and necessary adjustments and additions have been made to align the content with the standards and guidelines of your journal.

We sincerely appreciate your patient guidance and support throughout this process. Rest assured, we remain committed to maintaining the quality of the manuscript and eagerly anticipate any further feedback or guidance you may provide.

---

## [Decision Letter · Decision Letter 1]

6 May 2024

PONE-D-23-21385R1Effects of cardiac contractility modulation on autophagy and apoptosis of cardiac myocytes in rabbits with chronic heart failurePLOS ONE

Dear Dr. Liu,

Thank you for submitting your manuscript to PLOS ONE. After careful consideration, we feel that it has merit but does not fully meet PLOS ONE’s publication criteria as it currently stands. Therefore, we invite you to submit a revised version of the manuscript that addresses the points raised during the review process.

Your manuscript was reviewed by same reviewers and we received  positive feedback with few major comments. Please address all of them as appropriate during revision.

We look forward to receiving your revised manuscript.

Kind regards,

Partha Mukhopadhyay, Ph.D.

Section Editor

PLOS ONE

Journal Requirements:

Additional Editor Comments:

Reviewer's Responses to Questions

**Comments to the Author**

1. If the authors have adequately addressed your comments raised in a previous round of review and you feel that this manuscript is now acceptable for publication, you may indicate that here to bypass the “Comments to the Author” section, enter your conflict of interest statement in the “Confidential to Editor” section, and submit your "Accept" recommendation.

Reviewer #1: (No Response)

Reviewer #2: All comments have been addressed

2. Is the manuscript technically sound, and do the data support the conclusions?

Reviewer #1: Yes

Reviewer #2: Partly

3. Has the statistical analysis been performed appropriately and rigorously? 

Reviewer #1: Yes

Reviewer #2: Yes

4. Have the authors made all data underlying the findings in their manuscript fully available?

Reviewer #1: No

Reviewer #2: Yes

5. Is the manuscript presented in an intelligible fashion and written in standard English?

Reviewer #1: Yes

Reviewer #2: No

6. Review Comments to the Author

Reviewer #1: This manuscript by Hao et al. reports an interesting study on the application of cardiac contractility modulation (CCM) as an instrument treatment method in case of chronic heart failure. They showed that during heart failure in the rabbit animal model, CCM treatment can be implemented to improve/restore the systolic and diastolic behavior of the heart tissue (myocardium), which might be the result of the inhibition of the cardiomyocytes autophagy. In the revised version, the authors made some textual changes, which comparatively improve the quality of the manuscript. However, very surprisingly, there is no point-by-point response to the questionnaire that was requested by me (at least the revised version I am reading)! So, at this point, it is very difficult for me to make any decision.

JUST FOR THE RECORD, BELOW WERE SOME OF THE COMMENTS/QUESTIONS THAT I ASKED DURING THE 1ST ROUND OF REVIEW:

1. Authors provided a very short description of the chronic heart failure (CHF) and also mention the current/previous treatment methods (mostly beta-blocker) that have been attempted to treat this disease. However, for the general reader, it would be quite challenging to understand what is CHF and what happens during this phase. Moreover, it would be beneficial for the readers to briefly state the mode of action of the beta-blocker and how/why this method of treatment is less-effective (if it is). Why do the authors think that CCM can be a better strategy to treat CHF?

2. In the line 45-47, authors used a term “molecular modeling”. What does it mean? Molecular modeling of what? How is it happening? (Sorry, if I missed that info somehow!).

3. The methods section is well-written. However, materials section requires more details about the reagents and instruments that were used in these experiments.

4. The line 79 starts with “The slices were stored...”. Slices of what?

5. In figure 3, the authors checked the expression level of Bcl-2, Bax, Caspase-3 and LADH2 in Sham, HF and CCM groups and concluded that the expression of these mRNAs and the respective proteins in cardiomyocytes can reverse the development of cardiomyocytes apoptosis. Can authors also check the protein levels (by western blotting) from those differently expressed mRNA in Sham, HF and CCM groups and validate the above-mentioned statement?

6. Section 3.6 (line 176-180), authors looked at the cardiomyocytes apoptosis level by TUNEL assay, where they found that overall the apoptosis rate in HF and CCM group was increased compared with that in Sham group. Do authors have any justification of this observation? Also, is there any way to quantify this data (including the statistical significance) to get a better analysis of the apoptosis level?

7. Discussion requires more details and proper explanation of the results that authors presented in the manuscript. What is the molecular mechanism behind the effectiveness of CCM treatment in case of chronic heart failure? How does up regulation/down regulation of those proteins tested here in CCM group of animals justify this treatment mechanistically during chronic heart failure? In the discussion section, authors mentioned that previously they showed the positive effect of CCM in cardiac function, however, did not cite any of their publication. Why?

Overall, this study has merit and the experiments are well controlled and can be a nice addition to the research community of chronic heart failure. That being said, the manuscript requires additional details on the background, significance of this study and proper explanation of their data.

Reviewer #2: In the corrected manuscript, there still is some formatting and language issues. So I advise some minor alterations followed with a track changed resubmission.

7. PLOS authors have the option to publish the peer review history of their article (what does this mean?). If published, this will include your full peer review and any attached files.

Reviewer #1: No

Reviewer #2: No

---

## [Author Response · Author response to Decision Letter 1]

31 May 2024

Thank you for your comments, we have revised and responded to the previous questions and uploaded a file called "Point-to-point explanation" that I hope you can see.

---

## [Decision Letter · Decision Letter 2]

1 Aug 2024

PONE-D-23-21385R2

Effects of cardiac contractility modulation on autophagy and apoptosis of cardiac myocytes in rabbits with chronic heart failure

PLOS ONE

Dear Dr. Liu,

Thank you for submitting your manuscript to PLOS ONE. After careful consideration, we feel that it has merit but does not fully meet PLOS ONE’s publication criteria as it currently stands. Therefore, we invite you to submit a revised version of the manuscript that addresses the points raised during the review process.

*Comment from PLOS Office:* To comply with PLOS ONE submissions requirements, in your Methods section, please provide additional information regarding the experiments involving animals and ensure you have included details on (1) methods of sacrifice, (2) methods of anesthesia and/or analgesia, and (3) efforts to alleviate suffering, including information regarding humane endpoints. Specifically, please provide more details regarding the methods of sacrifice, as the current statement only reads "the animals in each group were euthanized with an overdose of anesthesia." Please also complete all items on the checklist at the following link: http://journals.plos.org/plosone/s/file?id=bb1d/plos-one-humane-endpoints-checklist.docx. Please upload the completed checklist as file type “Other” when resubmitting your manuscript. This document is for internal journal use only and will not be published if your article is accepted. We very much appreciate your attention to these requests and support of improved reporting standards in PLOS ONE submissions.

Additionally, in reviewing your submission we noticed you did not provide original raw image files supporting blot/gel data in the Figure 1A. We can see that you have provided cropped images of the membranes as supplementary file, however we require the original, uncropped and minimally adjusted images supporting all blot and gel results reported in an article’s figures and supporting information files (https://journals.plos.org/plosone/s/figures#loc-blot-and-gel-reporting-requirements). Can you please comment on this/these issue(s)?

When you reply, please also send the original raw image files (unadjusted, uncropped), clearly labeled and compliant with our guidelines at the above URL.

We look forward to receiving your revised manuscript.

Kind regards,

Johanna Pruller, PhD

Associate Editor

PLOS ONE

on behalf of

Partha Mukhopadhyay, Ph.D.

Section Editor

PLOS ONE

Journal Requirements:

Additional Editor Comments (if provided):

Reviewers' comments:

Reviewer's Responses to Questions

**Comments to the Author**

1. If the authors have adequately addressed your comments raised in a previous round of review and you feel that this manuscript is now acceptable for publication, you may indicate that here to bypass the “Comments to the Author” section, enter your conflict of interest statement in the “Confidential to Editor” section, and submit your "Accept" recommendation.

Reviewer #1: All comments have been addressed

Reviewer #2: All comments have been addressed

2. Is the manuscript technically sound, and do the data support the conclusions?

Reviewer #1: Yes

Reviewer #2: Yes

3. Has the statistical analysis been performed appropriately and rigorously? 

Reviewer #1: Yes

Reviewer #2: Yes

4. Have the authors made all data underlying the findings in their manuscript fully available?

Reviewer #1: Yes

Reviewer #2: Yes

5. Is the manuscript presented in an intelligible fashion and written in standard English?

Reviewer #1: Yes

Reviewer #2: Yes

6. Review Comments to the Author

Reviewer #1: The revised manuscript, titled “Effects of cardiac contractility modulation on autophagy and apoptosis of cardiac myocytes in rabbits with chronic heart failure,” investigated the application of cardiac contractility modulation (CCM) as an instrument treatment method in cases of chronic heart failure. The authors did a satisfying job of editing and revising the content of the manuscript. The authors have responded to all the questions I asked with reasonable and acceptable justification. Finally, the authors made nearly all the corrections and additions (textual, references, etc.) that were needed for the publication. Overall, this manuscript reads really well and would be a nice addition to the research community on chronic heart failure. I would recommend the publication of this manuscript once it satisfies the journal’s guidelines.

Reviewer #2: The authors have addressed the issues that were pointed out. The manuscript now matches the global standards of scientific reporting. This makes the topic to be depicted and understood in an intelligible manner.

7. PLOS authors have the option to publish the peer review history of their article (what does this mean?). If published, this will include your full peer review and any attached files.

Reviewer #1: No

Reviewer #2: No

---

## [Author Response · Author response to Decision Letter 2]

16 Sep 2024

Dear PLOS ONE，

Thank you for your valuable feedback and for accepting our request to conduct additional experiments. We have now completed the requested experiments and have revised the manuscript accordingly.

Additionally, we have included detailed descriptions of the procedures we used to minimize the pain and distress of the experimental animals. The revised manuscript has been re-uploaded for your consideration.

Please let us know if there are any further revisions or additional information needed.

Your Sincerely,

Huiliang Liu.

---

## [Editor Report · Decision Letter 3]

19 Sep 2024

Effects of cardiac contractility modulation on autophagy and apoptosis of cardiac myocytes in rabbits with chronic heart failure

PONE-D-23-21385R3

Dear Dr. Liu,

We’re pleased to inform you that your manuscript has been judged scientifically suitable for publication and will be formally accepted for publication once it meets all outstanding technical requirements.

Kind regards,

Partha Mukhopadhyay, Ph.D.

Section Editor

PLOS ONE
---

## [Editor Report · Acceptance letter]

2 Oct 2024

PONE-D-23-21385R3 

PLOS ONE

Dear Dr. Liu, 

I'm pleased to inform you that your manuscript has been deemed suitable for publication in PLOS ONE. Congratulations! Your manuscript is now being handed over to our production team.

Kind regards, 

on behalf of

Dr. Partha Mukhopadhyay 

Section Editor

PLOS ONE